# Lingering effects of COVID-19 on the mental health of first-year university students in Japan

**Ryo Horita**[1,2]*, **Akihiro Nishio**[1,2], **Mayumi Yamamoto**[1,2,3]

**1** Health Administration Center, Gifu University, Gifu, Japan, **2** Gifu University Hospital, Gifu, Japan, **3** United Graduate School of Drug Discovery and Medical Information Sciences, Gifu University, Gifu, Japan

* horita@gifu-u.ac.jp

## Abstract

### Background

The coronavirus disease 2019 (COVID-19) pandemic is continuing unabated in Japan, as of October 2021. We aimed to compare first-year university students' psychological distress before the pandemic in 2019, during the pandemic in 2020, and one year after the onset of the pandemic, in 2021.

### Methods

The study conducted online surveys over three years from April to May each year. Participants were 400 first-year students in 2019, 766 in 2020, and 738 in 2021. We examined differences in scores on the Counseling Center Assessment of Psychological Symptoms—Japanese version (CCAPS-Japanese) between the three years using a one-way analysis of variance, and differences in the CCAPS-Japanese critical items using chi-squared test and residual analysis.

### Results

The average scores on the Depression and Generalized Anxiety subscale in 2021 were significantly higher than those in 2020, but remained the same as in 2019. The Academic Distress subscale score in 2020 was the worst compared to 2019 and 2021. Meanwhile, the number of students who experienced severe suicidal ideation increased year by year from 2019 to 2021.

### Conclusion

The mean mental health of first-year university students worsened after the onset of the COVID-19 pandemic, and recovered to the pre-pandemic level over the next two years. However, the number of high-risk students with suicidal ideation continued to increase. A system is required for early detection and support for students at high risk of mental health issues.

**Data Availability Statement:** All relevant data are within the manuscript.

**Funding:** This work was supported by the Japan Society for the Promotion of Science KAKENHI Grant [grant number JP19K14446]. The funders

had no role in study design, data collection and analysis, decision to publish, or preparation of the manuscript.

**Competing interests:** The authors have declared that no competing interests exist.

## Introduction

How is the coronavirus disease 2019 (COVID-19) pandemic affecting the mental health of university students? A worldwide systematic Internet search showed that the suicide rate was unchanged or had declined in the early phase of the COVID-19 pandemic compared with the level in the pre-pandemic period in 21 high-income and upper-income countries, including Japan [1]. In contrast, the suicide mortality rates of graduate students in Japan were reported to slightly increase in an annual survey conducted by the Japanese National University Council of Health Administration [2]. Although the suicide mortality rate of undergraduate students in Japan has been on a downward trend since the 2013–2014 academic year, the 2020–2021 academic year had the highest rate in the last six academic years for men and in the last eight academic years for women [3]. It is speculated that the increased rates are due to the COVID-19 pandemic and not the long-standing culture of Japanese society. International studies on the impact of the COVID-19 pandemic on university students' mental health highlight its negative effects, such as increased depression and anxiety [4–7]. Studies on the impact of the pandemic in Asian countries indicated that international students had lower susceptibility to COVID-19 and higher anxiety than local students [8]. For example, Indian medical students experienced increased stress and anxiety due to being overburdened with responsibilities and a lack of adequate resources [9]. Another study on the psychological responses to COVID-19 among university students from three countries—Indonesia, Taiwan, and Thailand—revealed that Thai students had the highest levels of anxiety but the lowest levels of confidence in pandemic control and available resources for fighting COVID-19, and factors associated with high levels of anxiety differed across countries [10].On the other hand, we found that while first-year students in Gifu, Japan, experienced high academic distress in the year 2020, the number of "high-risk" students and those with depression were lower among first-year students in 2020 (i.e., during the pandemic) compared to the previous year (before the pandemic) [11].

The COVID-19 pandemic has continued unabated in Japan, similar to the situation in other developed countries. Gifu Prefecture, Japan, had declared a state of emergency at national and municipal levels four times by the end of May 2021, to combat the spread of COVID-19. From May 2020 to May 2021, the number of people infected with COVID-19 in Gifu Prefecture increased from 150 to 8,614, and in Japan, that level increased from 16,898 to 746,667. The emergency declaration promoted "The New Lifestyle," which consists of these basic rules: infection prevention, maintaining physical distance, wearing a mask, and washing hands. People refrained from going out unnecessarily and avoided the "Three C's" (crowded areas, closed spaces, and close-contact settings). Restaurants and pubs were closed or opened for short periods of time, and events were canceled or restricted in the number of people who could attend. As a consequence, the declaration imposed strict restrictions on the students for various campus activities, such as on-campus lectures, convivial meals with friends, travel, and club activities.

A cross-sectional study showed that subjective mental health status in 717 medical students significantly worsened after the nationwide state of emergency was declared. Those who had concerns about a shift toward online education had higher odds of having generalized anxiety and being depressed [12]. A web-based cross-sectional study revealed that approximately 70% of respondents had poor mental health [13]. They also indicated that having less communication with friends during self-quarantine was a risk factor for declining mental health. We can easily suspect that the adverse effects of the COVID-19 pandemic on the university students' mental health condition were caused by fear, self-isolation, physical distance, and so on. There are only a few cross-sectional studies on the mental health of health care college students or those in medical school during the COVID-19 pandemic in Japan [12–14]. Although the

results of only one cross-sectional mental health survey at a Japanese university, Akita Prefecture, were reported [15], there was no detailed analysis on how the COVID-19 pandemic affected the mental health of Japanese university students. In addition, most studies on COVID-19 and mental health have sampled university students from 2019 and 2020. Very few studies have considered 2021, thereby necessitating the current study.

This work extends our previous study [11] by comparing the new students' psychological distress before the pandemic in 2019, during the pandemic in 2020, and in 2021, one year after the onset of the pandemic. To detect and support early signs of maladjustment and mental crisis among new students, it is important to conduct surveys immediately after they first enroll in college. Therefore, the target sample of this study was first-year students. Our primary objective is to determine the lingering effects of COVID-19 on the mental health of first-year university students in Japan. Since we assume that the prolonged COVID-19 pandemic may have negatively impacted the mental health of university students, we postulate that the mental health of first-year university students in 2021 will be worse than in 2019 and 2020.

## Methods

### Study design

To examine the lingering effects of COVID-19, this study employed online surveys over three years, following the same procedure and instruments. Participants comprised students at Gifu University, a middle national university in Japan. We followed the STROBE guidelines when writing this paper [16].

### Participants and procedures

Participants comprised first-year Gifu University students in the 2019, 2020, and 2021 academic years. The total number of first-year students enrolled in Gifu University were 1,452 (19.8%) in 2019, 1,294 (17.6%) in 2020, and 1,266 (17.4%) in 2021. An online survey was conducted from April to May. All first-year students were informed and guided to fill out the questionnaire through email. For 24 hours, they could access the online survey from anywhere using their personal computers or smartphones and answer the questions by clicking the appropriate numbers. The participants were informed that the survey was unrelated to their academic evaluation, and they did not receive research participation credit.

In terms of demographic data, which included the biological sex and academic department of the students, the 2019 group consisted of 400 first-year students (226 female and 214 male students, response rate: 30.3%) from April 15 to May 31; the 2020 group consisted of 766 first-year students (347 female and 419 male students, response rate: 59.2%) from April 20 to May 31; and the 2021 group consisted of 738 first-year students (365 female and 373 male participants, response rate: 58.3%) from April 10 to May 31 (see also Table 1). During the survey period, the education methods of 2019, 2020, and 2021 were on-campus, e-learning, and hybrid (on-campus and e-learning), respectively.

### Measure

The study used the Counseling Center Assessment of Psychological Symptoms—Japanese (CCAPS-Japanese) [17,18] to measure university students' psychological symptoms over the previous two weeks. CCAPS is a multidimensional measure designed to assess psychological issues among college students [19]. Horita et al. [17,18] demonstrated the quality of the CCAPS-Japanese, which has rigorous factor structure, good internal consistency, adequate convergent validity, and good test-retest reliability. It comprises 55 items with eight factor-

**Table 1. Background characteristics of the participants in each group.**

| Student group | 2019 first-year (*N* = 440) | 2020 first-year (*N* = 766) | 2021 first-year (*N* = 738) |
|---|---|---|---|
| Survey period | from April 15 to May 31, 2019 | from April 20 to May 31, 2020 | from April 10 to May 31, 2021 |
| Measure | CCAPS-Japanese | CCAPS-Japanese | CCAPS-Japanese |
| Sex | | | |
| Female (%) | 226 (51.4) | 347 (45.3) | 365 (49.5) |
| Male (%) | 214 (48.6) | 419 (54.7) | 373 (50.5) |
| Academic department | | | |
| Education | 90 (20.5) | 156 (20.4) | 145 (19.6) |
| Reginal studies | 39 (8.9) | 80 (10.4) | 62 (8.4) |
| Medicine | 88 (20.0) | 115 (15.0) | 142 (19.2) |
| Engineering | 165 (37.5) | 277 (36.2) | 272 (36.9) |
| Applied biological sciences | 58 (13.2) | 138 (18.0) | 117 (15.9) |

*Note*. CCAPS-Japanese: The Counseling Center Assessment of Psychological Symptoms—Japanese.

derived subscales: Depression (11 items, e.g., "いつも悲しい ("I feel sad all the time")), Eating Concerns (8 items, e.g., "自分の体重に満足していない ("I am dissatisfied with my weight")), Hostility (7 items, e.g., "時々、何かを壊したい気持ちになる ("I sometimes feel like breaking or smashing things")), Social Anxiety (6 items, e.g., "知らない人といると居心地が悪い ("I feel uncomfortable around people I don't know")), Family Distress (6 items, e.g., "うちの家族には虐待があった ("There is a history of abuse in my family")), Alcohol Use (5 items, e.g., "飲酒が原因で後悔したことがある ("I have done something I have regretted because of drinking")), Generalized Anxiety (9 items, e.g., "色々な考えが次と浮かんでくる ("My thoughts are racing")), and Academic Distress (3 items, e.g., "学業についていけない ("I am unable to keep up with my schoolwork")), and four critical items: thought disturbance ("現実感がない ("I lose touch with reality")), suicidal ideation ("死にたいと考えることがある ("I have thoughts of ending my life")), violent behavior ("自制心を失い、暴力を振るうのではないかと心配である ("I'm afraid I may lose control and act violently")), and homicidal behavior ("暴力などで他人を傷つけることを考える ("I have thoughts of hurting others")). It is rated on a five-point Likert scale ranging from 0 (not at all like me) to 4 (extremely like me). Higher scores reflect higher levels of distress and symptoms. The Cronbach's α of the CCAPS-Japanese subscales in the 2019, 2020, and 2020 groups were as follows: Depression = .86–.88, Generalized Anxiety = .78–.83, Social Anxiety = .80–.82, Eating Concerns = .82–.83, Family Distress = .70–.77, Academic Distress = .60–.74, Hostility = .82–.84, and Alcohol Use = .72–.89.

## Defining variables and statistical analysis

Data were analyzed using the SPSS Statistics Ver. 27.0 (IBM Corp., Armonk, NY, USA). There were no missing data. We categorized participants who scored 0 on the critical items as "Low risk," scored 1 or 2 as "Moderate risk," and scored 3 or 4 as "High risk." Tests of differences in the eight CCAPS-Japanese subscales and the four CCAPS-Japanese critical items were investigated using one-way analysis of variance (ANOVA) for continuous variables, and chi-square and residual analysis for categorical variables. Multiple regression analysis was conducted to control for the potential confounders of sex and academic department. Independent samples *t*-test and ANOVA were conducted to examine differences due to the demographic factors of sex and academic department. Effect size based upon $\eta^2$ was examined to evaluate ANOVA results: small; $.01 \leqq \eta^2 < .06$, medium; $.06 \leqq \eta^2 < .14$, large; $.14 \leqq \eta^2$, and the effect size based

upon $d$ was examined to evaluate the $t$-test results based on the following criteria [20]: small, .20 $\leqq$ $d$ < .50; medium, .50 $\leqq$ $d$ < .80; and large, .80 $\leqq$ $d$. For the results of the residual analysis, we focused on cells with an absolute value of 1.96 (standard normal deviation of 5%) or higher for the adjusted standardized residual.

### Ethics statement

The research project was approved by the Research Ethical Committee of the Graduate School of Medicine, Gifu University, Japan (approval no. 28–320). All participants provided written informed consent to participate in the study. Further, they were notified that their responses would remain confidential and anonymous.

## Results

### Differences in the eight CCAPS-Japanese subscales

The one-way ANOVA showed the main effect of group on the CCAPS-Japanese Depression subscale score ($F$ (2, 1941) = 15.89, $p$ < .001, $\eta^2$ = .17). Post hoc analysis showed that the 2019 and 2021 first-year students (0.89±0.72, 0.88±0.70, respectively) were significantly higher on the Depression subscale results than the 2020 students (0.71±0.61) ($ps$ < .001). Further, the one-way ANOVA also showed the main effect of group on the CCAPS-Japanese Generalized Anxiety subscale ($F$ (2, 1941) = 8.30, $p$ < .001, $\eta^2$ = .09). Post hoc analysis showed that the 2019 and 2021 first-year students (1.02±0.63, 1.05±0.69, respectively) had significantly higher scores on the Generalized Anxiety subscale than the 2020 students (0.92±0.62) ($p$ < .05, $p$ < .001, respectively).

In contrast, one-way ANOVA showed the main effect of group on the CCAPS-Japanese Academic Distress subscale ($F$ (2, 1941) = 35.24, $p$ < .001, $\eta^2$ = .36). Post hoc analysis showed that the 2019 and 2021 first-year students (1.23±0.77, 1.15±0.71, respectively) were significantly lower on the Academic Distress subscale results than the 2020 students (1.47±0.79) ($ps$ < .001).

However, no significant differences were found in the mean levels of any other CCAPS-Japanese subscale between 2019, 2020, and 2021 (Table 2).

**Table 2. Results of comparing the CCAPS-Japanese subscales for 2019, 2020, and 2021 first-year students.**

| | First-year students in 2019 (N = 440) | | First-year students in 2020 (N = 766) | | First-year students in 2021 (N = 738) | | F | Effect Size $\eta^2$ |
|---|---|---|---|---|---|---|---|---|
| | **M** | **SD** | **M** | **SD** | **M** | **SD** | | |
| **CCAPS-Japanese subscale** | | | | | | | | |
| Depression | 0.89 | 0.72 | 0.71 | 0.61 | 0.88 | 0.70 | 15.89*** | 0.17 |
| Generalized Anxiety | 1.02 | 0.63 | 0.92 | 0.62 | 1.05 | 0.69 | 8.30*** | 0.09 |
| Social Anxiety | 2.01 | 0.89 | 1.89 | 0.88 | 1.91 | 0.91 | 2.53 | 0.03 |
| Eating Concerns | 0.92 | 0.70 | 0.95 | 0.71 | 0.97 | 0.72 | 0.85 | 0.01 |
| Family Distress | 0.67 | 0.66 | 0.68 | 0.58 | 0.71 | 0.64 | 0.63 | 0.01 |
| Academic Distress | 1.23 | 0.77 | 1.47 | 0.79 | 1.15 | 0.71 | 35.24*** | 0.36 |
| Hostility | 0.63 | 0.65 | 0.59 | 0.62 | 0.60 | 0.66 | 0.49 | 0.01 |
| Alcohol Use | 0.03 | 0.17 | 0.04 | 0.17 | 0.02 | 0.17 | 1.14 | 0.02 |

*Note*. $M$, mean; $SD$, standard deviation; CCAPS-Japanese, Counseling Center Assessment of Psychological Symptoms—Japanese, $\eta^2$ = Effect size, small; .01$\leqq\eta^2$ < .06, medium; .06$\leqq\eta^2$ < .14, large; .14$\leqq\eta^2$. Statistically significant differences were analyzed using the analysis of variance.

*$p$ < .05,
**$p$ < .01,
***$p$ < .001.

## Correlation of demographic characteristics and the eight CCAPS-Japanese subscales

The correlation between the demographic characteristics of sex and academic department and the eight CCAPS-Japanese subscales was analyzed by multiple regression analysis. There were significant correlations between sex and the Eating Concerns subscale among the 2019, 2020, and 2021 first-year students, and the Alcohol Use subscale among the 2020 first-year students. There were also significant correlations between academic department and the Social Anxiety and Academic Distress subscales in 2020 first-year students.

Independent samples $t$-tests revealed that scores on the Eating Concerns subscale were significantly higher for female students than male students among first-year students in 2019 ($t$ (420.17) = 5.46, $p < .001$, $d = .52$), 2020 ($t$ (640.47) = 6.75, $p < .001$, $d = .49$), and 2021 ($t$ (674.98) = 6.82, $p < .001$, $d = .50$); and scores on the Alcohol Use subscale were significantly lower for female students than male students among first-year students in 2020 ($t$ (496.40) = 4.18, $p < .001$, $d = .30$). An ANOVA revealed that scores on the Social Anxiety subscale were significantly lower for 2020 first-year students in the education department than those in the applied biological sciences department ($F$ (4, 761) = 3.17, $p < .05$, $\eta^2 = .02$), and scores on the Academic Distress subscale were significantly lower among 2020 first-year students in the education department than those in the engineering department ($F$ (4, 761) = 3.36, $p < .01$, $\eta^2 = .02$). Since there were no significant correlations between sex or academic department and Depression, Generalized Anxiety, Family Distress, or Hostility, only the significant differences are presented in Tables 3 and 4.

## Differences in the four CCAPS-Japanese critical items

Chi-square analyses on each of the four critical items revealed differences for only two items: "I lose touch with reality" (thought disturbance), $\chi^2$ (4) = 18.26, $p < .001$; and "I have thoughts of ending my life" (suicidal ideation), $\chi^2$ (4) = 11.13, $p < .05$. To determine which cells were significantly different, a residual analysis was performed.

Residual analysis of thought disturbance revealed that the number of low-risk students was significantly higher among the 2019 group (n = 284, 64.5%, adjusted standardized residual (asr) = 3.44), whereas it was lower among the 2020 group (n = 415, 54.2%, asr = -2.32). In addition, the number of moderate-risk students was significantly lower among the 2019 group (n = 115, 26.1%, asr = -3.55), and the number of high-risk students was significantly higher

**Table 3. Sex differences in the CCAPS-Japanese subscales.**

|  | female | | male | | $t$ | Effect Size $d$ |
|---|---|---|---|---|---|---|
|  | *M* | *SD* | *M* | *SD* |  |  |
| **CCAPS-Japanese subscale** |  |  |  |  |  |  |
| Eating Concerns in 2019 | 1.09 | 0.75 | 0.74 | 0.58 | 5.46*** | 0.52 |
| Eating Concerns in 2020 | 1.14 | 0.78 | 0.79 | 0.60 | 6.75*** | 0.49 |
| Eating Concerns in 2021 | 1.16 | 0.79 | 0.81 | 0.59 | 6.82*** | 0.50 |
| Alcohol Use in 2020 | 0.01 | 0.06 | 0.06 | 0.22 | 4.18*** | 0.30 |

*Note. M*, mean; *SD*, standard deviation; CCAPS-Japanese, Counseling Center Assessment of Psychological Symptoms—Japanese, $d$ = Effect size, small; $.20 \leqq d < .50$, medium; $.50 \leqq d < .80$, large; $.80 \leqq d$. Statistically significant differences were analyzed using the independent samples t-tests.

*$p < .05$,

**$p < .01$,

***$p < .001$.

**Table 4. Differences by academic department in the CCAPS-Japanese subscales.**

|  | Education | | Regional studies | | Medicine | | Engineering | | Applied biological sciences | | *F* | Effect Size η² |
|---|---|---|---|---|---|---|---|---|---|---|---|---|
|  | *M* | *SD* | *M* | *SD* | *M* | *SD* | *M* | *SD* | *M* | *SD* |  |  |
| **CCAPS-Japanese subscale** |  |  |  |  |  |  |  |  |  |  |  |  |
| Social Anxiety in 2020 | 1.75 | 0.89 | 2.03 | 0.80 | 1.80 | 0.92 | 1.89 | 0.84 | 2.06 | 0.90 | 3.17* | 0.02 |
| Academic Distress in 2020 | 1.33 | 0.77 | 1.35 | 0.73 | 1.39 | 0.75 | 1.57 | 0.79 | 1.54 | 0.84 | 3.36** | 0.02 |

*Note. M*, mean; *SD*, standard deviation; CCAPS-Japanese, Counseling Center Assessment of Psychological Symptoms—Japanese, η² = Effect size, small; $.01 \leqq η^2 < .06$, medium; $.06 \leqq η^2 < .14$, large; $.14 \leqq η^2$. Statistically significant differences were analyzed using the analysis of variance.

*$p < .05$,

**$p < .01$,

***$p < .001$.

among the 2020 group (n = 86, 11.2%, asr = 2.14), whereas it was lower among the 2021 group (n = 57, 7.7%, asr = -2.05).

Residual analysis of suicidal ideation revealed that the number of low-risk students was significantly higher among the 2020 group (n = 571, 74.5%, asr = 2.79). and the number of moderate-risk students was significantly higher among the 2019 group (n = 120, 27.3%, asr = 2.09). However, it was lower in the 2020 group (n = 162, 21.1%, asr = -2.02), and the number of high-risk students was significantly higher among the 2021 group (n = 50, 6.8%, asr = 2.01) (Table 5).

## Discussion

The main purpose of this study was to examine the lingering effect of COVID-19 on Japanese first-year university students' mental health. Drawing on the assertions by Horita et al. [11], the present research compared three points in time of students' mental health. We predicted that the mental health of 2021 first-year university students would be the worst of the three because of the prolonged pandemic. Although the response rates varied across the three time points, the ratios by sex and academic department were similar to those of Gifu University at all three time points. Therefore, it can be assumed that there was no selection bias.

One of the interesting findings in this study was that the depression and anxiety levels of 2020 first-year students were the highest, while those in 2021 were at the same level as in 2019. In other words, the mean mental health of first-year students had returned to the pre-pandemic level over the past two years. In contrast, the number of low-risk 2020 students was high, whereas that of high-risk 2021 students was high in suicidal ideation. This is consistent with the findings of Ueda et al. [21], which suggested that the number of suicide deaths during the initial phase of the pandemic was lower than average but exceeded the past trend from July 2020. The data for the 2020 first-year students in this study were from May 2020. The number of students who scored the maximum value (4) for the suicidal ideation item increased each year (2019: n = 3, 0.68%, 2020: n = 8, 1.04%, 2021: n = 12, 1.63%). According to Japan's National Police Agency [22], the number of deaths by suicide in 2020 reached 415 among university students, which was 25 more than that in 2019. Although the speculation was not supported in terms of the mean of students' mental health, the number of students with severe mental health problems may be increasing. Thus, universities should establish a system of early detection and provide support for students, especially those who struggle with severe mental distress. Fushimi [23] reported that the number of students having mental health

**Table 5. Results of comparing the CCAPS–Japanese critical items for 2019, 2020, and 2021 first-year students.**

| | | First-year students in 2019 | First-year students in 2020 | First-year students in 2021 |
|---|---|---|---|---|
| **Thought disturbance** | | | | |
| Low risk | n (%) | 284 (64.5) | 415 (54.2) | 417 (56.5) |
| | asr | 3.44 | -2.32 | -0.63 |
| Moderate risk | n (%) | 115 (26.1) | 265 (34.6) | 264 (35.8) |
| | asr | -3.55 | 1.11 | 1.94 |
| High risk | n (%) | 41 (9.3) | 86 (11.2) | 57 (7.7) |
| | asr | -0.12 | 2.14 | -2.05 |
| **Suicidal ideation** | | | | |
| Low risk | n (%) | 297 (67.5) | 571 (74.5) | 512 (69.4) |
| | asr | -1.83 | 2.79 | -1.22 |
| Moderate risk | n (%) | 120 (27.3) | 162 (21.1) | 176 (23.8) |
| | asr | 2.09 | -2.02 | 0.23 |
| High risk | n (%) | 23 (5.2) | 33 (4.3) | 50 (6.8) |
| | asr | -0.24 | -1.79 | 2.01 |
| **Violent behavior** | | | | |
| Low risk | n (%) | 376 (85.5) | 667 (87.1) | 626 (84.8) |
| | asr | -0.27 | 1.25 | -1.02 |
| Moderate risk | n (%) | 55 (12.5) | 89 (11.6) | 95 (12.9) |
| | asr | 0.15 | -0.73 | 0.61 |
| High risk | n (%) | 9 (2.0) | 10 (1.3) | 17 (2.3) |
| | asr | 0.34 | -1.24 | 1.16 |
| **Homicidal behavior** | | | | |
| Low risk | n (%) | 383 (87.0) | 669 (87.3) | 641 (86.9) |
| | asr | -0.03 | 0.26 | -0.24 |
| Moderate risk | n (%) | 45 (10.2) | 78 (10.2) | 75 (10.2) |
| | asr | 0.03 | 0.00 | -0.03 |
| High risk | n (%) | 12 (2.7) | 19 (2.5) | 22 (3.0) |
| | asr | 0.00 | -0.54 | 0.54 |

*Note*. Low risk: Scored 0, moderate risk: Scored 1 or 2, high risk: Scored 3 or 4, and asr: Adjusted standardized residual; statistically significant differences were analyzed with residual analysis.

consultations was higher than normal, especially soon after the beginning of the second semester in 2020. He suspected that the stress and mental health of university students might change from the early stage to the next stage of the pandemic and future crises.

By contrast, 2021 students experienced lower academic distress than 2020 students. We assumed that academic distress would progressively worsen according to the prolonged pandemic; however, this finding was contrary to our predictions. The 2020 students had to prepare for and adapt to an unfamiliar e-learning environment [11]. It is likely that the 2021 students were familiarized with the new learning environment during the 2020 academic year when they were in high school. Conversely, the pressure of online courses may appear only after a month or two, when students find that the classes can be disorganized despite not having to leave their homes and saving traffic time. Moreover, they might feel that online classes are more relaxed due to less social stress.

The relationship between demographic factors and distress was also examined. For eating concerns, female students were significantly more distressed than male students at all three time points. Since excessive awareness of food and physical appearance can lead to eating

disorders, it is necessary to provide effective support for eating habits, especially for female students. Alcohol use in 2020 was significantly higher among male students than among female students. However, most of the participants in this study were under 20 years old, and students of this age are prohibited by law from drinking alcohol. Therefore, more empirical evidence seems warranted before we reach substantive conclusions about the relationship between COVID-19 and drinking habits. The study also needs to be replicated with the over 20 years old population.

With regard to thought disturbance (critical item), while 2020 first-year students might feel detached from the outside world when staying home for two months due to the pandemic [11], this trend was mitigated for the 2021 first-year students. A plausible reason for our finding could be that university students might be promptly accepting the changes in their lives caused by COVID-19 and adapting to "The New Lifestyle," as well as adapting to changes in their learning environment. Hatabu, Mao, Zhou, Kawashita, Wen, Ueda, et al. [24] demonstrated that the overall knowledge, attitudes, and practices toward COVID-19 among university students in Japan were high. All respondents showed they possessed knowledge of the "Three C's." Most respondents showed a moderate or higher frequency of washing their hands or wearing masks (both at 96.4%). In addition, 68.5% of the respondents showed a positive attitude toward early drug administration.

This study is the first to compare the mental health of first-year students in Japanese universities before the pandemic, during the pandemic, and one year after the onset of the pandemic, using the CCAPS. Since the CCAPS was developed as an instrument used in research to assess national and global trends in university students' mental health, only this study has shown specific mental health problems occurring in higher education related to the COVID-19 pandemic. There are only four cross-sectional survey reports on university students' mental health during the pandemic in Japanese universities: medical students using nine questions [12], health care college students using the GHQ-12 and self-reports [13], medical students using the K-6, Rosenberg Self-Esteem Scale, and General Self-Efficacy Scale [14], and national university students in Akita using the PHQ-9 [15] (see Table 6). Roughly, they showed a worsened mental health condition with increased anxiety and depression after a state of emergency. The unique contribution of this study is that we reveal the lingering effects of COVID-19 on the mental health of first-year students by comparing data from three points in time. We showed that decreased depression

**Table 6. Comparison of the surveys.**

|  | Nishimura et al. | Tahara et al. | Arima et al. | Nomura et al. | Our survey |
|---|---|---|---|---|---|
| Participants | medical students | health care college students | medical students | national university students in Akita | first-year national university students in Gifu |
| Survey year | 2020 | 2020 | 2020 | 2020 | 2019, 2020, and 2021 |
| n | 717 | 223 | 571 | 2712 | 1944 (440 in 2019, 766 in 2020, and 738 in 2021) |
| Measures | 9 questions | GHQ-12 and self-report | K-6, Rosenberg Self-Esteem Scale, and General Self-Efficacy Scale | PHQ-9 | CCAPS-Japanese |
| Main results | The participants' subjective mental health status significantly worsened after the state of emergency | Less communication with friends was the main risk factor for mental health problem. Good health status and satisfaction with leisure and new activities were associated with reduced risk of mental health problem. | In the students with a K-6 score ≥5, higher scores on Self-Esteem correlated with lower levels of psychological distress, whereas those with higher Self-Efficacy scores also scored higher for indicators of psychological distress. | Negative lifestyles of smoking and drinking, and being a woman, may be important risk factors for depressive symptoms, whereas exercise and having someone to consult about worries may be protective factors. | The mean of 2021 first-year students' depression and anxiety had returned to pre-pandemic of the COVID-19. The number of students with severe mental health problems may be increasing. The 2021 first-year students felt lower academic distress than in 2020. |

and generalized anxiety as well as increased academic distress in the first phase of the pandemic had recovered to the level before the pandemic after almost one year of the pandemic.

## Limitations

There are several limitations concerning the research methodology. First, this study provides a snapshot of mental health, but does not show whether it is prolonged or tapers off. Second, our results are limited to university students enrolled in a local city. Such a selection may introduce an element of selection bias in the study. Therefore, the results need to be verified in other regions and sizes of populations for the generalization of findings. Based on these limitations, more empirical evidence is warranted before we reach more definitive conclusions about the effects of COVID-19 on the mental health of first-year university students in Japan.

## Future directions

There are two future directions for research in this area. First, future research might focus on the impact of lifestyle, behavioral characteristics, and psychological properties on mental health among first-year students. Nomura et al. [15] suggested that smoking, drinking, and being a woman may be important risk factors for depressive symptoms, whereas exercise and having someone to consult about worries may be protective factors among Japanese university students during the COVID-19 stay-at-home order. Marashi et al. [25] find that mental health problems play a dual role—they act as barriers to physical activity while motivating physical activity during the COVID-19 pandemic. Among students with mental health problems due to COVID-19, students with high self-esteem showed lower levels of psychological distress, while students with high self-efficacy showed higher levels of psychological distress [14]. Tahara et al. [13] showed that loneliness could be the major reason for the decline in the mental health of students, and participation in leisure and new activities is a proactive strategy that supports good mental health during self-quarantine due to COVID-19. Feng, Zong, Yang, Gu, Dong, and Qiao [26] demonstrated that individuals with high altruism exhibited more negative affect than those with low altruism, which indirectly increased their anxiety and depressive symptoms. In addition to the above, factors such as residence form (family home, student dormitory, or living alone) and college enrollment process (going directly onto a university or after failing to enter a university) may also have an impact on the mental health of first-year students who have just enrolled in a university. The results of this study indicated that students in the education department had slightly lower levels of stress than students in other departments, which suggests that there may be differences in emotional perception by academic department. Second, the COVID-19 pandemic in Japan should be ending in the near future. Another interesting question is what the mental health of first-year students will be after the cessation of COVID-19.

## Conclusion

The unique contribution of this study is that we compare data between three points in time to demonstrate the prolonged effect of COVID-19 on Japanese first-year university students' mental health. The average score of mental health declined after the COVID-19 pandemic, and returned to the pre-pandemic level over the next two years. However, the number of students at high risk of mental health issues continued to increase, and universities should establish a system for early detection and support for such students.

## Supporting information

**S1 Appendix. CCAPS-Japanese questionnaire.**
(PDF)

## Acknowledgments

We would like to thank Editage (www.editage.com) for English language editing.

## Author Contributions

**Conceptualization:** Ryo Horita.

**Data curation:** Ryo Horita, Akihiro Nishio, Mayumi Yamamoto.

**Formal analysis:** Ryo Horita.

**Funding acquisition:** Ryo Horita.

**Investigation:** Ryo Horita.

**Methodology:** Ryo Horita.

**Project administration:** Ryo Horita.

**Resources:** Ryo Horita.

**Supervision:** Mayumi Yamamoto.

**Writing – original draft:** Ryo Horita.

**Writing – review & editing:** Ryo Horita, Akihiro Nishio, Mayumi Yamamoto.

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
