## [Decision Letter · Decision Letter 0]

22 Nov 2021

PONE-D-21-31952Lingering effects of COVID-19 on the mental health of first-year university students in JapanPLOS ONE

Dear Dr. Horita,

Thank you for submitting your manuscript to PLOS ONE. After careful consideration, we feel that it has merit but does not fully meet PLOS ONE’s publication criteria as it currently stands. Therefore, we invite you to submit a revised version of the manuscript that addresses the points raised during the review process.

One expert in this filed has reviewed this work and you will see her detailed comments below. I agree with her opinions that your paper has merits and values during the COVID-19 pandemic periods; however, something  (e.g., using regression models to control potential confounders) should be fixed before going forward. Therefore, please carefully respond to her comments for improving your work. Apart from her comments, I personally feel that your paper should integrate more COVID-19 references on university students' mental health. Please see the suggestions below.Sharma R, Bansal P, Chhabra M, Bansal C, Arora M. Severe acute respiratory syndrome coronavirus-2-associated perceived stress and anxiety among Indian medical students: A cross-sectional study. Asian J Soc Health Behav 2021;4:98-104Ahorsu, D. K., Pramukti, I., Strong C., Wnag, H.-W., Griffiths, M. D., Lin, C.-Y., Ko, N.-Y. (2021). COVID-19-related variables and its association with anxiety and suicidal ideation: Differences between international and local university students in Taiwan. Psychology Research and Behavior Management, 14, 1857-1866. Pramukti, I., Strong, C., Sitthimongkol, Y., Setiawan, A., Pandin M. G. R., Yen, C.-F., Lin, C.-Y., Griffiths, M. D., Ko, N.-Y. (2020). Anxiety and suicidal thoughts during the COVID-19 pandemic: A cross-country comparison among Indonesian, Taiwanese, and Thai university students. Journal of Medical Internet Research, 22(12), e24487. In addition, please provide the entire CCAPS-Japanese questionnaire as a supplementary file. Regarding the psychometrics of the CCAPS-Japanese (i.e., the comment raised by the reviewer), you can use the present sample to conduct (e.g., Cronbach's alpha or confirmatory factor analysis). ==============================

We look forward to receiving your revised manuscript.

Kind regards,

Chung-Ying Lin

Academic Editor

PLOS ONE

Reviewers' comments:

Reviewer's Responses to Questions

**Comments to the Author**

1. Is the manuscript technically sound, and do the data support the conclusions?

Reviewer #1: Partly

2. Has the statistical analysis been performed appropriately and rigorously? 

Reviewer #1: No

3. Have the authors made all data underlying the findings in their manuscript fully available?

Reviewer #1: No

4. Is the manuscript presented in an intelligible fashion and written in standard English?

Reviewer #1: Yes

5. Review Comments to the Author

Reviewer #1: This is a valuable study. During the COVID-19 pandemic, the authors investigated the anxiety and depression of first-year college students for three years in an attempt to see and explain the impact of the COVID-19 pandemic.

The following are my comments:

Introduction

Please explain why it is specifically targeted at first-year college students rather than university students as a whole.

Please describe the change in the proportion of anxiety, depression, suicide, violence and other tendencies across the country over the five years before 2019 in order to confirm that the change in numbers was the effect of the COVID-19 pandemic, not the long-standing culture of other societies.

Methods

The authors are recommended to use regression analysis to control confounding factors, for example, gender, department, and family income.

Please provide more information about psychometric properties of CCAPS-Japanese.

Results

Please present the student's department composition in order to reduce the interference of the department with emotional perception.

Line 163 I wonder why the author thought η2 = .09 was very small. According to your criteria, 0.09 was medium.

Please add the percentages after the number of students.

Discussion

Please discuss the impact of differences in annual response rates on results. In addition, what is the proportion of the whole school to fill in the questionnaire? How is the questionnaire sent to everyone? Was there a selection bias?

Line 212-213 I am confused that the worst depression indicating more depressed or less?

For students, the pressure of online courses may only appear in a month or two, after which students will find that classes can be mixed up without going out to save a lot of traffic time. Moreover, they might feel that the line class industry more relaxed due to less social stress.

6. PLOS authors have the option to publish the peer review history of their article (what does this mean?). If published, this will include your full peer review and any attached files.

Reviewer #1: No

---

## [Author Response · Author response to Decision Letter 0]

26 Dec 2021

Response to Reviewer 1:

I deeply appreciate your supportive comments. I have edited my previous manuscript, and my responses to your suggestions are as follows. I have also highlighted the corresponding text in the manuscript.

Introduction

Comment:

Please explain why it is specifically targeted at first-year college students rather than university students as a whole. Please describe the change in the proportion of anxiety, depression, suicide, violence and other tendencies across the country over the five years before 2019 in order to confirm that the change in numbers was the effect of the COVID-19 pandemic, not the long-standing culture of other societies.

Response: Per your suggestion, I explained why the study is specifically targeted at first-year university students (P. 6 L. 97–99).

In Japan, there is no national survey on anxiety, depression, and violence. Because I only have access to national surveys on suicide, we described the change in the suicide rates (P. 4 L. 49–53).

Methods

Comment:

The authors are recommended to use regression analysis to control confounding factors, for example, gender, department, and family income. Please provide more information about psychometric properties of CCAPS-Japanese.

Response: Thank you for your kind advice. I have conducted multiple regression analysis using biological sex and department, and the results are shown in Tables 3 and 4 in the “Correlation of demographic characteristics and the eight CCAPS-Japanese subscales” and “Discussion” section (P. 17 L. 305–P. 18 L. 314 and P. 21 L. 374–377). We did not ask the participants about family income. 

We have also described more precisely the description of the CCAPS (P. 8 L. 138–141) and submitted the entire CCAPS-Japanese questionnaire as a supplementary file. 

Results

Comment:

Please present the student's department composition in order to reduce the interference of the department with emotional perception. Line 163 I wonder why the author thought Η2 = .09 was very small. According to your criteria, 0.09 was medium. Please add the percentages after the number of students. 

Response: Thank you for your helpful comments. I have presented the composition of the students by academic department in Table 1. 

Per your suggestion, I have modified the description of the effect size evaluation.

Based on your suggestion, I have demonstrated the percentage data as well as numbers in Table 5.

Discussion

Comment:

Please discuss the impact of differences in annual response rates on results. In addition, what is the proportion of the whole school to fill in the questionnaire? How is the questionnaire sent to everyone? Was there a selection bias? Line 212-213 I am confused that the worst depression indicating more depressed or less? For students, the pressure of online courses may only appear in a month or two, after which students will find that classes can be mixed up without going out to save a lot of traffic time. Moreover, they might feel that the line class industry more relaxed due to less social stress.

Response: I appreciate your suggestion. Although there was a difference in the response rate among the three time points, there were no significant differences in the breakdown by sex or academic department in each group, and these percentages were similar to those of Gifu University as a whole. In addition, there was no intention to access the survey. Therefore, I determined that our study did not have any selection bias (P. 16 L. 274–276). 

The ratio of first-year students to all students is shown in the “Methods” section (P. 6 L. 115––P. 7 L. 116). 

Instructions on how students were asked to submit the questionnaire were included in the “Methods” section (P. 7 L. 116–121). 

I apologize for the confusion regarding the wording for depression. I have now clearly stated, “highest scores” for anxiety and depression (P. 16 L. 278). 

I edited the discussion regarding online classes as you suggested (P. 17 L. 301–304).

---

## [Editor Report · Decision Letter 1]

28 Dec 2021

Lingering effects of COVID-19 on the mental health of first-year university students in Japan

PONE-D-21-31952R1

Dear Dr. Horita,

We’re pleased to inform you that your manuscript has been judged scientifically suitable for publication and will be formally accepted for publication once it meets all outstanding technical requirements.

Kind regards,

Chung-Ying Lin

Academic Editor

PLOS ONE
---

## [Editor Report · Acceptance letter]

4 Jan 2022

PONE-D-21-31952R1 

Lingering effects of COVID-19 on the mental health of first-year university students in Japan 

Dear Dr. Horita:

I'm pleased to inform you that your manuscript has been deemed suitable for publication in PLOS ONE. Congratulations! Your manuscript is now with our production department. 

Kind regards, 

on behalf of

Dr. Chung-Ying Lin 

Academic Editor

PLOS ONE